# What happens to Fracture Energy in Brittle Fracture?

# Revisiting the Griffith Assumption.

Timothy R. H. Davies[1], Maurice J. McSaveney[2], Natalya V. Reznichenko[1]

[1]Department of Geological Sciences, University of Canterbury, Christchurch 8140, New Zealand

5  [2]GNS Science Ltd, PO Box 30368 Lower Hutt, New Zealand

*Correspondence to*: Tim Davies (tim.davies@canterbury.ac.nz)

**Abstract.** Laboratory experiments involving unconfined compressive failure of borosilicate glass cylinders quantified the elastic strain energy released at failure and the size distribution of the resulting fragments. The data were carefully assessed for potential inaccuracies in surface-area calculation, of the contribution of energy from the compression machine relaxation during specimen failure, and of possible variations in the specific fracture energy of the specimens. The data showed that more new surface area was created during the failures than would be possible if the long-standing assumption, that all the energy involved in creating new rock surface area in brittle material is taken up by the newly-created surfaces as surface potential energy and is not available to do further work, were valid. We therefore conclude that the assumption is false. This conclusion is supported by independent data from a previous investigation whose authors did not pursue this particular application. Our result does not affect the validity of Griffith fracture mechanics, and is significant only when large numbers of very fine fragments are created by brittle fracture, as in rock-avalanche motion and earthquake rupture, and are identified in particle-size distributions. In such situations our result is very significant to understanding of fracture energetics.

# 1 Introduction

The energy transformations that occur during brittle fracture are extremely important in a wide range of phenomena, such as landslides, earthquakes, structural failures and rock crushing for aggregate production. For example, in a rock avalanche the kinetic energy developed in the fall of a rock mass from its initial source location to its final deposit location is the source of all the forces that resist friction during motion, including the forces that create the intense rock fragmentation resulting in the creation of new rock surface area. Thus one might expect that the greater the new surface area created by fragmentation, the shorter might be the runout (e.g. Hungr, 2006).

An expression for the energy required to extend a crack in brittle material is presented in Griffith (1921). It provides an energy balance to calculate the total potential (elastic strain) energy in a perfect specimen body uniformly stressed, then estimates the amount by which it is reduced when a crack of known dimensions is introduced; the reduction in energy is the energy required to create the crack, which scales with the new surface area created on both sides of the crack. The specific fracture energy U ($Jm^{-2}$) is then a property of the material. Griffith (1921) states as if it were fact, that the energy used to create the crack is transformed to "surface potential energy", whose character is not detailed but is assumed to play no further part in the process. Since the energy balance explains the relationship between crack area and the energy input required in forming it, the later fate of this energy is irrelevant to the derivation and whether it becomes unavailable or remains available does not affect the crack propagation. The statement that it

becomes unavailable for doing further work is an untested assumption that has persisted ever since, and has rarely been questioned (McSaveney & Davies, 2009). It is important because, for example, in the context of rock avalanches mentioned above, it implies that the energy lost in creation of new surface ought to result in reduced rock-avalanche runout because it is not available to overcome friction. Hence

intensely-fragmented debris should be associated with shorter runout distances; but the opposite is true (e.g. Davies, 2018).

The present work addresses the validity of the assumption that the energy used in the creation of new surface area by brittle fracture becomes unavailable for further work by being sequestered as surface potential energy. We do NOT in any way question the validity of the relationship between the reduction

of elastic energy and the area of new surface created; that is, we do not question the validity of linear elastic fracture mechanics, only the fate of the energy used to create new surface

Recent developments in the technology for measuring the sizes of fine particles now allow accurate measurement of sub-micron particle-size distributions. Individual fragments as small as a few nanometers (nm; $10^{-9}$ m) can now be detected, and size distributions can be quantified to a few tens of nm (Chester et

al., 2005). Associated with these ultra-fine particles are very large specific surface areas (up to 80 $m^2g^{-1}$; e.g. Wilson et al., 2005, Barber & Griffith, 2018). Creating these large areas of new surface requires

corresponding large energy inputs, and in Earth surface processes this energy must come from the potential (gravitational and/or tectonic) energy driving the process.

Recently Davies et al. (in press) found that the gravity potential energy available for emplacement of the Lake Coleridge rock avalanche deposit in New Zealand was only just sufficient to supply the energy lost

5   to friction in that event, leaving only about 2% of the total potential energy available to generate the large quantity of new surface area created; this, however, should have required about 70% of the potential energy. These field data called into question the validity of the long-standing assumption due to Griffith (1921), that all the energy used to create new rock surface in brittle fracture immediately transforms into surface potential energy that is not available to do further mechanical work.

10   The present work reports a laboratory test of this question, by analyzing the fragment surface area resulting from failure of Pyrex cylinders in unconfined compression. Here the total energy input to the system is measured, and compared with the total energy required to create the measured surface area of the fragments, including those down to about 70 nm in diameter. The data show unequivocally that the energy input to the Pyrex cylinders is insufficient to explain the total area of new surface produced during the

fragmentation, if all fracture energy is lost to surface energy. Our laboratory data are supported by independent data from Kolzenberg et al. (2013), who however did not develop this aspect of their data.

These test results suggest that the conventional Griffith assumption, that all fracture energy becomes surface energy, is false. This is of great significance to the understanding of fracture situations that generate large quantities of fine fragments.

## 2 Brittle fracture: Griffith theory

Classical Griffith fracture theory provides a simple explanation of the stress requirements for a crack to enlarge: a crack will propagate when the forces holding two sides of a potential crack together are exceeded by forces pulling them apart (Griffith, 1921; Lamb, 1995). This concept was presented mathematically as an energy balance at the tip of an incipient crack in an ideal brittle material, but it is significant that the mathematics of the energy balance do not involve the later history of the energy provided to extend the crack – in other words, what happens to the energy once the crack extension is complete. Nevertheless, when a crack enlarges in otherwise intact material, new surface area is created, and Griffith (1921) stated that the energy used to create it (to which we refer herein as fracture surface energy, FSE) then becomes "surface potential energy" (Gibbs, 1873), which is associated with the new surface and not available for doing further work on the system. Henceforth we refer to surface potential energy as surface free energy (SFE), following modern usage (Chibowski et al., 1989; Zgura et al., 2013; Savvova et al., 2015). Irwin (1957) modified the original Griffith brittle failure hypothesis by including plasticity in the region of the crack tip, which led to Griffith's work being accepted as the foundation of

modern brittle failure theory. In most applications of Griffith brittle failure theory, FSE is treated as an energy sink (e.g. Miller et al., 1999; Chester et al., 2005; Hungr, 2006; Grady, 2008; Livne et al, 2010), on the assumption that it transforms completely to SFE.

However, the Griffith (1921) statement, that the energy involved in propagating a crack is (entirely) lost to potential energy residing on the pair of new surfaces created by the extension of the crack, is in fact an assumption. Whether or not this assumption is true has no effect on the validity of the Griffith fracture theory, which only addresses the mathematical requirements for a crack to enlarge – i.e. the value of FSE. The theory is not about SFE and says nothing about the complex chemical processes associated with exposure of fresh material by enlargement of cracks.  Nevertheless the assumption that FSE is completely lost to SFE has endured for almost a century, and has rarely been specifically questioned hitherto (McSaveney & Davies, 2009).

## 3 Experiments

### 3.1 Overview

Our experiments involved unconfined compression to failure of 20-mm diameter by 40-mm long cylinders of borosilicate glass ("Pyrex"). The failures were generally catastrophic and generated large numbers of fragments down to sub-micron scale. By carefully monitoring the load on and strain of the sample to failure, and analysing in detail the resulting particle-size distribution, we were able to relate the energy available to create new surface to the area of new surface created. We point out that whether the fragments were produced in the primary failure of the cylinder, or as a result of collision of a high-velocity

fragment with the container, is immaterial; in the latter case the fragment kinetic energy also derived from the strain energy released at failure. If the energy-loss assumption of Griffith (1921) were valid, the new surface area would be limited by the known specific surface energy of Pyrex (i.e. the energy required to generate 1 sq. m of new surface; this is 4.5 $Jm^{-2}$, e.g. Wiederhorn, 1969: Lange, 1971). However we measured much more new surface than this constraint allows. Data from similar experiments by Kolzenberg et al. (2013) yielded the same result, although those authors did not develop this particular interpretation of their data.

This result is only apparent because very large numbers of extremely small fragments were generated *and measured* in brittle fracture. Analysis of sub-micron fragments has only recently become possible, so that the large proportion of total surface area associated with this fraction can now be identified and quantified. An additional issue in this measurement is the fact that the finest fragments generated can bond together to form much larger agglomerates (Reznichenko et al., 2012) immediately they are created, and these must be disaggregated to reveal the full fragment size distribution generated by fragmentation.

## 3.2 Experimental Procedure

Our experimental design was based on that of Kolzenberg et al. (2013), using identically-sized cylinders of the same Pyrex material. However, our experiments took place in unconfined conditions while those of Kolzenberg et al. (2013) were carried out with the Pyrex cylinder confined in a latex sheath and subject to a range of confining pressures applied by argon gas. Failure loads and deflections were similar in both test series, as would be expected from the identical sample sizes and materials (Table 1).

Our Pyrex cylinders were accurately machined to have parallel ends, and the ends were coated with a molybdenum sulphide-based lubricant to limit stress build-up at the ends during longitudinal compression. Cylinders were annealed to remove any residual internal stress by heating to 1000˚C for 15 minutes and cooling to room temperature over 24 hours, before being set between steel platens in an oversized metal cylinder that prevented any fragments escaping. The cylinder contained air at atmospheric pressure, room temperature and ambient humidity. Samples were compressed longitudinally and uniaxially in a Tecnotest Compression Testing Machine Model KE300/ECE at a constant stress

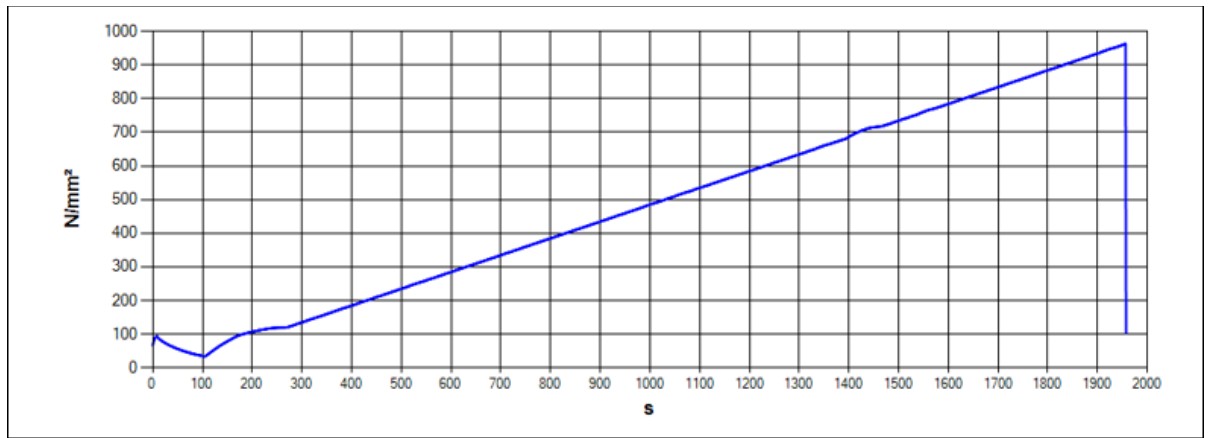

Fig. 1 Stress (N/mm$^2$) against time (s) for test UC1

application rate of 0.50 MPa/s, and time to failure was of the order of 15-30 minutes; this machine is marketed as having a stiff frame. Stress-strain curves were strongly linear except for some deviation at very low load (e.g. Fig. 1). Data (load and deflection) needed to calculate strain energy input were logged by the compression machine.

### 3.3 Machine strain energy and specimen strain energy

The machine compressing the specimen also stores strain energy, which is released at specimen failure and can contribute to the energy available to generate new surface area during the failure of the specimen. Here we calculate the magnitude of this contribution.

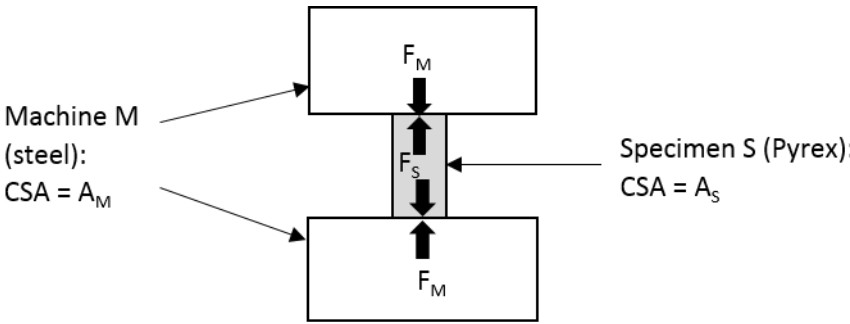

Machine M (steel): CSA = $A_M$

Specimen S (Pyrex): CSA = $A_S$

Fig. 2    Definition diagram for estimating machine strain energy

Under elastic conditions the forces acting on the machine and on the specimen balance (Fig. 2):

$$F_M = F_S = F. \tag{1}$$

The compressive stress in the machine $\sigma_M = F/A_M$, and the compressive stress in the specimen $\sigma_S = F/A_S$; thus the strain in the machine $\delta_M = \sigma_M/E_M$, and the strain in the sample $\delta_S = \sigma_S/E_S$.

The elasticity of the machine (steel) $E_M \approx 3$ x the elasticity of the specimen (Pyrex) $E_S$. The cross-sectional area of the machine $A_M$ is greater than the cross-sectional area of the specimen $A_S$ (however this is defined, e.g. total volume/longitudinal dimension).

The strain energy of the machine $SE_M = 0.5\ F\delta_M = 0.5F(F/A_M E_M)$; the strain energy of the specimen $SE_S$

$= 0.5F\delta_S = 0.5F(F/A_S E_S)$. So

$$\frac{SE_S}{SE_M} = \frac{0.5F(\frac{F}{A_S E_S})}{0.5F(\frac{F}{A_M E_m})} = \frac{A_M E_M}{A_S E_S} = 3\,{}^{A_M}/_{A_S} \qquad (2)$$

Hence, since $A_M > A_S$, $SE_S > 3SE_M$; and the strain energy stored in the machine is less than 1/3 that stored in the specimen.

While the whole of the machine frame stores elastic strain energy during specimen compression, only that released during specimen failure can contribute to the energy available to create new fragment surface. The celerity of an elastic wave in steel is about the same as in Pyrex – about 5400 m/sec. Hence an elastic wave traverses the 40 mm long Pyrex cylinder in $0.04/5400 = 7.4\ \mu sec$, which is the maximum time for complete failure of the cylinder; while an elastic wave traverses the $> 200$ mm long steel machine

in $> 0.2/5400 = 37\ \mu sec$, which is the time taken for full relaxation of the machine and transmission of all its strain energy to the specimen. Thus $< 7.4/37 = < 20\%$ of machine strain energy, which is less than

about 6% of total strain energy, can contribute to cylinder fragmentation, assuming steady release of machine strain energy.

We conclude that strain energy released from the testing machine upon specimen failure contributes a small quantity of strain energy to the specimen during failure. In the context of the present data this proportion can be neglected without affecting the conclusions.

## 3.4 Particle-size distributions

The fragments resulting from cylinder failure were collected (about 98% by weight recovery, in part due to glass fragments remaining embedded in the steel containment vessel), the sample was dry-sieved and the fraction passing a 63-micron sieve separated. About 5 gm of the sub-63-micron fraction was put into a container and submerged in ethanol, then put for 10-15 minutes into an ultrasonic bath at maximum energy setting in order to disperse any agglomerates that may have formed during or following fragmentation (Reznichenko et al., 2012), and was then decanted into a long tube containing ethanol and allowed to settle for 24 hours. We show later that there was no possibility that the ultrasound exposure could have formed new cracks, or extended existing cracks, to form additional new surface beyond that created in the failure of the sample.

The settled sample was removed from the settling tube by pipette and introduced to the Saturn Digisizer 5200 laser analysis equipment that was full of distilled water. The flow rate was set to maximum, and the ultrasonic probe run at maximum intensity for 60 seconds. Data acquisition took place under prescribed

conditions, and analysis (Fraunhofer) involved the refractive index (1.474) and density (2230 kg/m$^3$) of Pyrex. Analyses were re-run multiple times on each sample.

Particle-size distributions from the laser analysis are shown in Fig. 3. It is notable that UC 1, which failed at the highest stress of 917 MPa, generated a higher proportion of fine fragments than the other three UC
samples. It is also notable that all of the UC samples had detectable proportions of fragments as fine as 0.4 microns; by comparison, the Kolzenberg et al. (2013) samples produced no fragments smaller than 0.83 microns, which may be related to their confinement in latex jackets.

## 3.5 Surface-area calculations

From the particle-size data, we calculated the cumulative surface area for each size fraction by assuming
firstly that all fragments were spherical; then we applied a correction factor k to account for the fact that the fragments were in fact angular (Fig. 4) so had larger surface area per unit mass than spheres. Wilson et al. (2005) measured the ratio of surface areas between angular and spherical grains in fault gouge as 6.6, and Chester et al. (2005), Hochella and Banfield (1995) and White et al. (1996) state more generally that this ratio usually lies in the range 5-10 for angular debris from rock fracture. However, we require a
k value specifically for our Pyrex fragments; we did this by analyzing the fragment shapes in three SEM images (one of which is Fig. 4), showing a total of over 500 fragments.

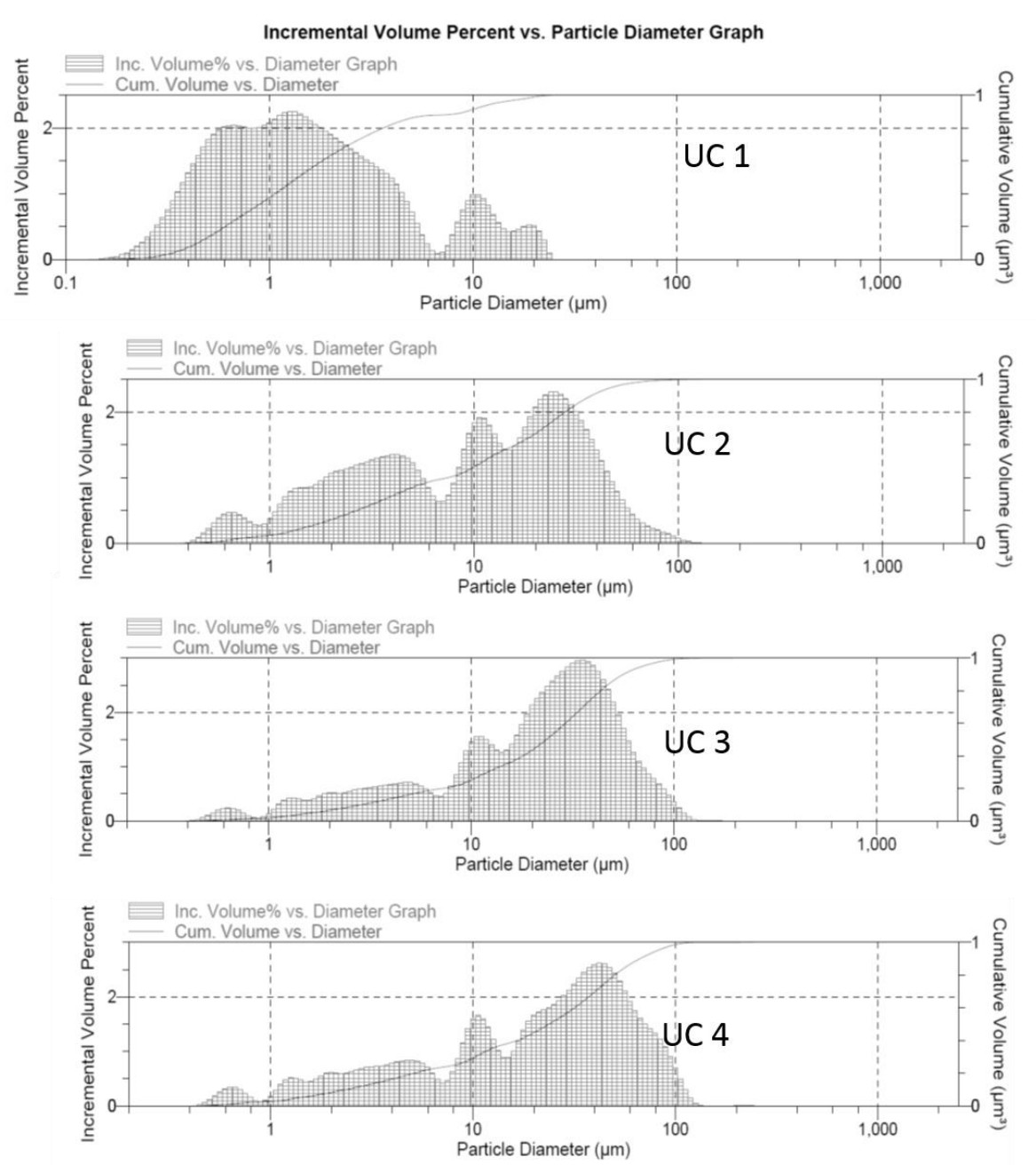

Fig. 3 Laser particle-size data for fragmented Pyrex from present tests

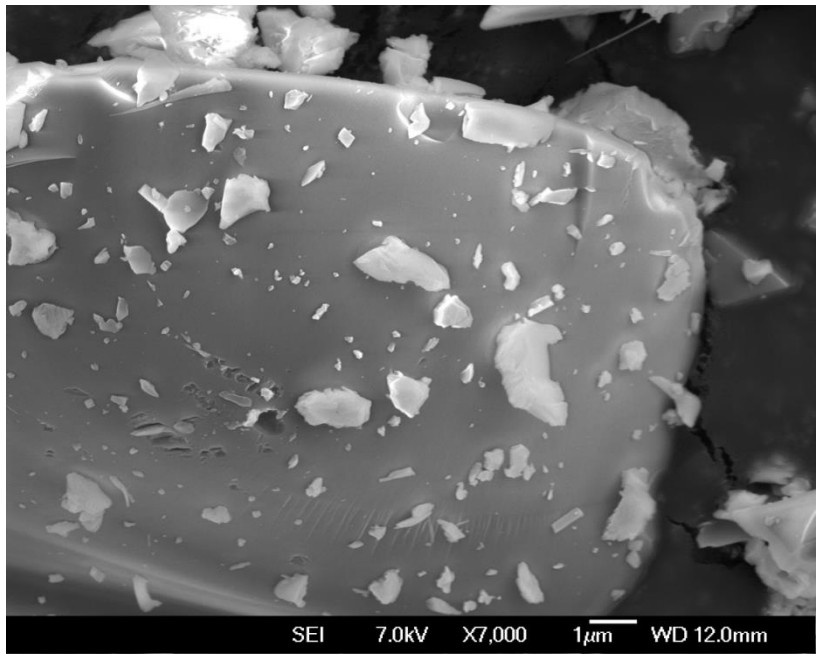

Fig. 4   SEM image of fragments from Pyrex failure

First we derived from SEM images of Pyrex fragments (e.g. Fig. 4) the ratio of the fragment image perimeter to the perimeter of the "equivalent circle" (the circle with the same area as the fragment).

5   Next we considered two regular shapes: a rectangle (2D) of height d and length nd, and a square-section bar (3D) of height and width d and length nd. For n between 1 and 10, we calculated the ratio of rectangle perimeter to equivalent circle perimeter, and of bar surface area to equivalent sphere (the sphere with the same volume as the bar) surface area. For each value of n we calculated the factor K that relates the latter ratio to the former; this turned out to be close to 3 for n = 10 and about 3.45 for n = 1.

We then multiplied the ratio of fragment perimeter to equivalent circle perimeter, from over 500 fragment images, by K = 3 (in order to be conservative) to give an estimate of fragment surface area to equivalent sphere surface area. In greater detail:

Figure 5 illustrates a three-dimensional fragment and the equivalent sphere (whose diameter is output by the laser-sizing apparatus).

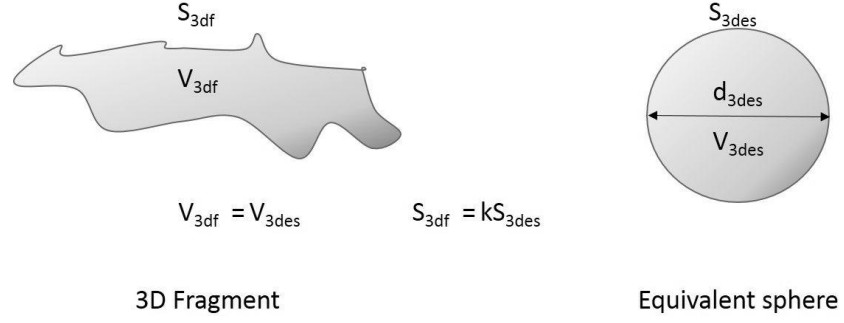

Figure 5. Three-dimensional fragment and equivalent sphere

From SEM imagery we can measure the ratio of (2D) fragment perimeter to the perimeter of the equivalent circle (circle of equal area) for fragments in our tests $P_{2df}/P_{2dec}$; P = perimeter, A = area (Fig. 6).

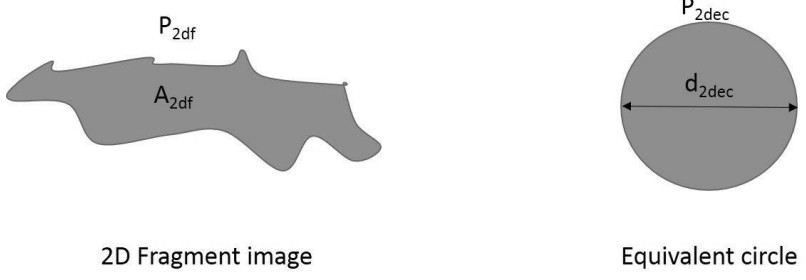

Figure 6. Two-dimensional fragment image and equivalent circle.

We need to calculate $k = S_{3df}/S_{3des}$ from known values of $P_{2df}/P_{2dec}$. We now derive the relationship between these ratios for regular 2D and 3D shapes (Fig. 7).

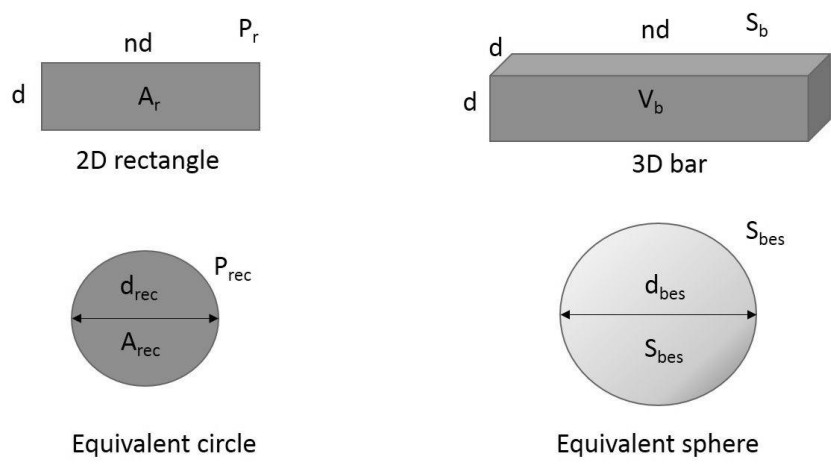

Figure 7. Two-dimensional rectangle with equivalent circle (left); three-dimensional square-section bar with equivalent sphere (right)

For regular shapes as in Fig. 7, we assume $S_b/S_{bes} = K(P_r/P_{rec})$; then it can be shown geometrically that

$$\frac{P_r}{P_{rec}} = \frac{(1+n)}{(n\pi)^{0.5}} \quad and \quad \frac{S_b}{S_{bes}} = \frac{2(1+2n)}{\pi^{1/3}(6n)^{2/3}} \tag{3}$$

Table 1 shows values of $P_r/P_{rec}$, $S_b/S_{bes}$ and K for $1 \leq n \leq 10$

| n | $S_b/S_{bes}$ | $P_r/P_{rec}$ | K |
|---|---|---|---|
| 1 | 6.9 | 2 | 3.45 |
| 2 | 11.5 | 3.36 | 3.415 |
| 3 | 16.1 | 4.80 | 3.351 |
| 4 | 20.7 | 6.29 | 3.285 |
| 5 | 25.3 | 7.84 | 3.224 |
| 6 | 29.9 | 9.43 | 3.168 |
| 8 | 39.1 | 12.72 | 3.071 |
| 9 | 43.7 | 14.42 | 3.029 |
| 10 | 48.3 | 16.14 | 2.991 |

Table 1.   $P_r/P_{rec}$,  $S_b/S_{bes}$ and K for $1 \leq n \leq 10$.

If we assume that

$$\left(\frac{S_b}{S_{bes}}\right)\left(\frac{P_r}{P_{rec}}\right) = K = \left(\frac{S_{3df}}{S_{3des}}\right)/\left(\frac{P_{2df}}{P_{2dec}}\right) \tag{4}$$

Then

$$\left(\frac{S_{3df}}{S_{3des}}\right) = k = K\left(\frac{P_{2df}}{P_{2dec}}\right) \tag{5}$$

and k can be calculated from SEM images. For each of three images we obtain the results in Table 2.

| Number of fragments in image | $P_{2df}/P_{2dec}$ max | k maximum | k average |
|---|---|---|---|
| 144 | 2.378 | 7.136 | 3.9 |
| 285 | 2.925 | 8.776 | 4.1 |
| 105 | 2.628 | 7.886 | 4.1 |

Table 2.    k values derived from fragment SEM images

The average k values in Table 2 are close to 4. These will be underestimates because our calculation assumes that there is no irregularity of the fragments in the dimension not shown in the images, suggesting

increasing the k values by 33%. Further, the maximum values of k in Table 2 are somewhat less than 10, so our analysis suggests that our k values are close to the range 5-10 as found empirically by other investigators. Thus the present analysis supports the value of k = 5 used in the estimation of fragment specific surface area.

Kolzenberg et al. (2013) found by SEM imagery that most of their fragments were approximately cubic in shape, perhaps due to the confinement of their samples in latex sheaths under non-zero confining pressures; they calculated their fragment surface areas accordingly.

Finally, we corrected the fragment surface area for the fraction (< 63 µm) of the fragments of the whole cylinder debris (< 5 mm) that were measured in the laser-sizer, which was about 25%. The calculated

total surface areas of the Pyrex cylinder debris are listed in Table 3, where they are compared with the theoretical surface able to be created if the energy used to create each square metre of surface then becomes unavailable.

## 4. Data

The experimental data from our tests and those of Kolzenberg et al. (2013) are listed in Table 3 and

illustrated in Fig. 8. Table 3 and Fig. 8 show that in all cases more new surface area was created than would be possible if the Griffith (1921) assumption were valid (i.e. all the elastic strain energy used to create new surface area (FSE) became surface free energy (SFE)). Even if corrections for machine energy (+6% available energy) and k value (4 instead of 5; -20% surface area), which were neglected above, are

applied, data point UC3 (the closest to the limit line) has surface area 9.9 m$^2$ and available energy 41.2 J, compared to the 44.6 J required by the Griffith assumption, so it still shows an energy deficit. In the extreme case where the machine energy is 1/3 of the energy stored in the specimen and is added to the specimen energy, there is still insufficient energy to create the surface area measured in all except two cases (UC3 and UC4). The total accumulated errors are shown as vertical lines descending from the data points in Fig. 8. Thus the possible inaccuracies in our data do not affect the overall result.

Similarly the effect of possible variations in the specific fracture energy of borosilicate glass ($4.5 \pm 0.22$ Jm$^{-2}$; Wiederhorn, 1969) are included in the location of the dashed line in Fig. 8 and have negligible effect. The theoretical maximum surface area corresponding to the energy available are also estimated (Table 1 final column) assuming that the specific fracture energy is 4.28 Jm$^{-2}$.

The data in Table 3 and Fig. 8 indicate that up to 5 times more surface area was created than allowed by the Griffith assumption; this suggests strongly that the assumption is not valid.

Our (UC) data were obtained in air at ambient humidity, whereas the data of Kolzenberg et al. (2013) were obtained in a dry (argon) atmosphere. It is known that the presence of water vapour reduces the specific fracture energy of Pyrex (Wiederhorn, 1969), and this could affect our data; however it is difficult to envisage how moisture in the outside atmosphere could affect the interior of an impermeable (non-crystalline) cylinder. Thus the *new* surface would have been generated under moisture-free conditions in our experiments, to which the specific surface energy of 4.5 Jm$^{-2}$ applies. The two sets of data correspond well in the present context, supporting this contention.

| Sample name | Confining pressure MPa | Peak Stress MPa | Stress Drop MPa | Post-failure strength MPa | Total energy J | Surface area m² | Theoretical surface area m² (U = 4.28) |
|---|---|---|---|---|---|---|---|
| **Kolz. 1** | 25 | 1277 | 1251 | 26 | 148.5 | 127.52 | 34.70 |
| **Kolz. 2** | 50 | 1046 | 842 | 204 | 145.7 | 88.16 | 34.04 |
| **Kolz. 3** | 75 | 1051 | 756 | 295 | 161.3 | 59.32 | 37.69 |
| **Kolz. 4** | 100 | 1293 | 761 | 532 | 179.9 | 54.88 | 42.03 |
| **Kolz. 6** | 15 | 1389 | 1372 | 18 | 146.3 | 184.22 | 34.18 |
| **Kolz. 7** | 5 | 835 | 825 | 10 | 75.5 | 97.66 | 17.64 |
| **Kolz. 8** | 0.1 | 648 | 645 | 3 | 52.3 | 67.9 | 12.22 |
| **UC 1** | 0 | 917 | 917 | 0 | 133.2 | 91.8 | 31.12 |
| **UC 2** | 0 | 606 | 606 | 0 | 46.6 | 22.4 | 10.89 |
| **UC 3** | 0 | 500 | 500 | 0 | 38.9 | 12.4 | 9.09 |
| **UC 4** | 0 | 667 | 667 | 0 | 43.7 | 14.2 | 10.21 |

Table 3 Pyrex fragmentation data from Kolzenberg et al. (2013) (Kolz.) and from the present study (UC)

We assume that ultrasonic treatment only disaggregates particles that are weakly bonded to each other, and is not able to fracture intact glass. In order to test this assumption we subjected 500-micron glass
5  beads to ultrasound at the same intensity and for the same duration as the Pyrex fragments; we found no difference in size distribution between pre-and post-ultrasound analysis and SEM examination confirmed

lack of breakage of glass beads. Since the ultrasonic energy density required to break particles is inversely related to their size (Knoop et al., 2016), we concluded that ultrasound treatment did not cause breakage of intact micron-scale Pyrex fragments, only disaggregation of previously-agglomerated or previously-cracked grains.

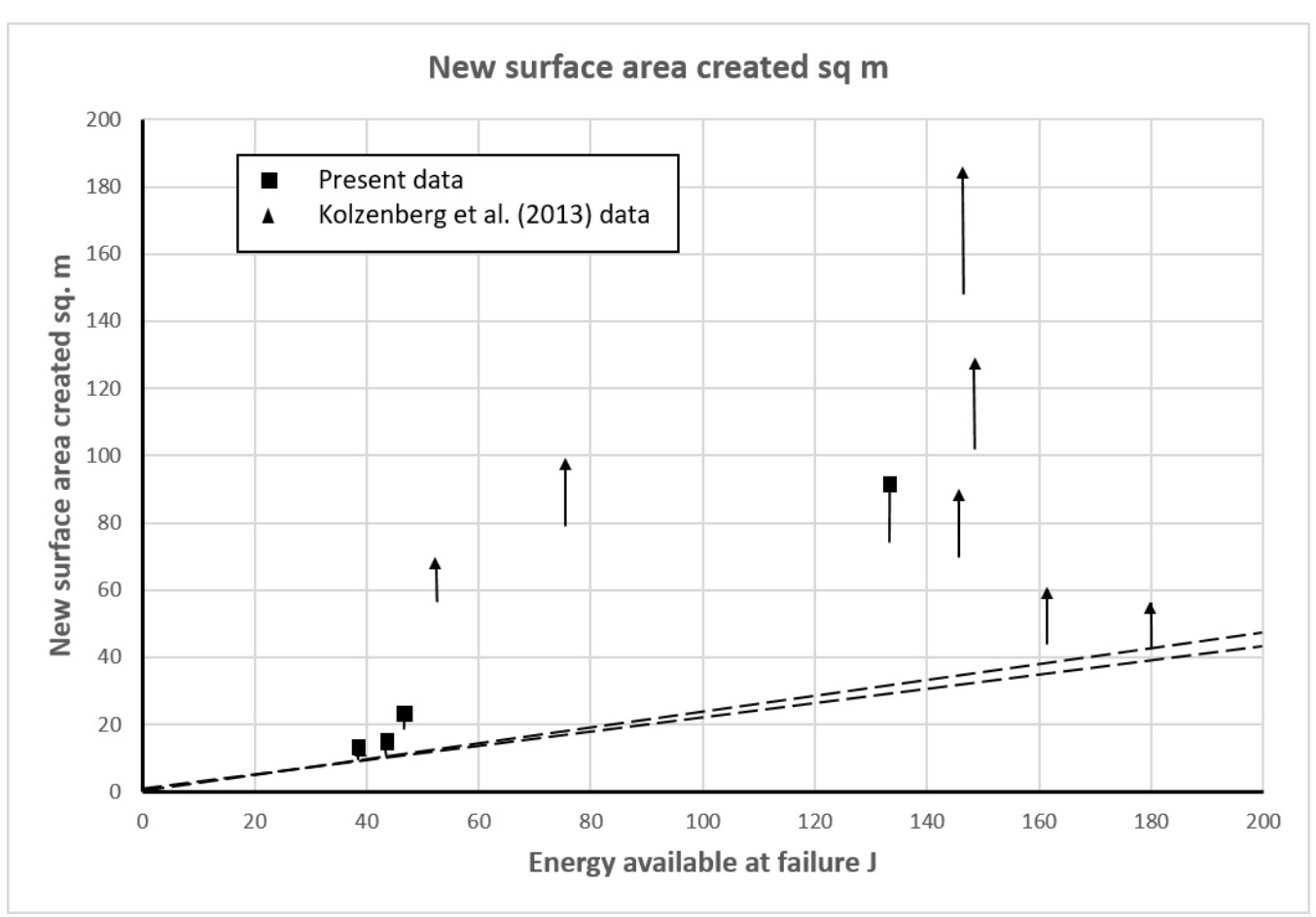

Fig. 8 Experimental data from the present work (squares) and from Kolzenberg et al.(2013) (triangles) showing the new surface created by failures involving various energies. Vertical lines descending from data points indicate extreme possible negative errors. The dashed lines show the limits on surface area imposed by the specific surface energy of Pyrex ($4.5 \pm 0.22$ Jm$^{-2}$) if the energy used for fracture is lost to surface energy. It is notable that all the data points lie above these lines, and some are far above. The maximum theoretical strain energy that can be stored in Pyrex is $Q^2/2E$ Jm$^{-3}$ (Herget, 1988; Q is the unconfined compressive strength of the material and E is its elasticity); this is about 200 J in the experimental cylinders.

## 5. Fate of fracture energy

The data in Fig. 8 require that some of the elastic strain energy that had caused crack extension remained available to drive further crack growth, because there was no other source for the energy to create the new surface area in excess of that used to create the theoretical surface area. If this were not the case, substantially less new surface would necessarily have been created. Eventually, of course, the energy involved in a fragmentation event decays by friction and noise to heat and surface free energy; our concern herein is the form that the elastic strain energy released by cylinder failure took before it became heat and ceased to be available for further work. Recent descriptions of brittle fracture processes lack clarity as to the fate of the energy involved in creating new surface. Most simply state that all of the fracture surface energy is (immediately) "consumed" as surface free energy, following Griffith (1921); but some make different assumptions. For example, Gudmunsson (2011) states that the energy used to break molecular

bonds in splitting a rock clast "...is absorbed in the solid body..." of the fragments, but without specifying in what form it is absorbed. This statement indicates that the Griffith assumption, while it is the common convention, is not universally accepted.

To resolve this conundrum we start by clearly restating the terminology involved in fracture energetics.

1) The energy required to break intermolecular bonds and form new surface area is the *Fracture Surface Energy* (FSE); this is equal to the failure strength of an individual bond, multiplied by the number of bonds per unit area of surface. It is calculable theoretically and measurable by careful fracture experiments; for pure quartz it is close to 2 $Jm^{-2}$ (Ball and Payne, 1976), while for crustal rocks in general it lies between 7-200 $Jm^{-2}$ (Friedman et al, 1972; Chelidze et al, 1994; Ouchterlony, 1982). The empirical value for borosilicate glass is 4.5 $Jm^{-2}$ (Wiederhorn, 1969: Lange, 1971).

2) The energy associated with a new rock surface is the *Surface Free Energy* (SFE). It is the energy per unit area of (broken) intermolecular bonds. It represents the potential of the surface to interact chemically with the medium adjacent to it, so is chemical energy. When bonds break much of their stress is released, and one would expect their residual energy to be substantially less than the energy associated with a bond that is strained to breaking point. This is borne out by empirical data: Savvoya et al. (2015) find a surface free energy of 54.1 mJ/sq m for borosilicate glass using contact angle techniques. This is 1.2% of FSE. Chibowski et al. (1988) found by two independent methods that the polar surface free energy component of glass is $80 \pm 16$ $mJ/m^2$, while the

dispersion surface free energy component was $29 \pm 9$ mJ/m$^2$. Investigating quartz, Zgura et al. 2013 found that: "The surface energy of natural or synthetic quartz and silica films and particles was described by a large range of values between 50 and 230 mN/m and the literature is rich from this point of view…"; Zdziennicka et al. (2009) reported surface free energy values for quartz in the range 176-206 mN/m. Hence SFE is about 4-11% of FSE in quartz, which unlike Pyrex is crystalline. In general, SFE appears to be a small proportion of FSE.

The statements of Griffith (1921) that the energy per unit area required to extend a brittle fracture (FSE) "...appears as potential surface energy..." (i.e. SFE), and that "...the total decrease in potential energy due to the formation of a crack is equal to the increase in strain energy less the increase in surface energy", are sufficiently imprecise to have allowed the universal adoption of the conventional interpretation that *all* the FSE transforms to SFE. However, the numerical difference between FSE and SFE demonstrated above shows that this cannot be the case.

It appears from the above that, at most, *some* of the fracture surface energy can transform to surface free energy as a crack propagates and new surface forms. If only 1% of FSE transforms to SFE in breakage of Pyrex, as the empirical data of Savvova et al. (2015) suggest, then the theoretical limit to generation of new surface area in Table 3 is 100 times the value in the final column – up to almost 4000 m$^2$. This clearly resolves the difficulty with the conventional assumption that our data highlight.

On this basis, we suggest that a large proportion of the energy used to fracture rock (i.e. to create and extend cracks in intact rock) is *absorbed into the solid fragments* (as Gudmunsson, 2011 suggested) *as elastic body-wave energy* – that is, as high-frequency vibrations – and hence remains available to do further work (such as in causing further fragmentation). This should not be counted as an energy loss. McSaveney and Davies (2009) arrived at the same conclusion from a detailed consideration of the nature of surface energy. Livne et al. (2010) visualized energy flowing towards a crack-tip as it extends; they, however, assumed that this energy "dissipated" (presumably to surface energy) as the crack extended. By contrast, we suggest instead that most of this energy radiates as body-wave energy, in line with Svetlizky et al. (2016). Thus we picture a single intact grain in a shearing mass deforming elastically under shear and storing elastic strain energy as it does so; upon exceeding its brittle yield strength the grain becomes many individual fragments, all of which immediately rebound to their unstrained shapes (subject to changing confinement in the grain flow), releasing most of the stored elastic strain energy and delivering it to adjacent grains as elastic body-wave energy. A small proportion of the released elastic strain energy transforms to become the surface free energy of the new surface.

## 6. Discussion

While our data were acquired under specific conditions of system geometry, material and applied stress configuration, which may perhaps have been particularly favourable for reaching our outcome, our demonstration that the Griffith assumption is not valid in these circumstances is a very strong indicator

that it is invalid in all circumstances. Griffith's (1921) original assumption was associated with a very general derivation of his theory, and therefore implies general applicability of both theory and assumption.

While the data presented are widely scattered (Fig. 8), this is irrelevant to the specific objective of our work, which is to demonstrate that the new surface area generated in brittle fracture is not limited to the

zone below the dashed lines in Fig. 8, as it would be if the Griffith assumption were valid.

We note that we have ignored the possibility that plastic deformation occurred in the glass failure. Irwin (1958) assumed that irreversible plastic deformation can occur. If in cracking of Pyrex glass if there were crushed-but-not-disaggregated grains, or microfractures in the surviving fragments, they would represent additional fracture surface area created that would not be accounted for in our calculation of surface area

from the particle-size distributions. That is, if there is microcracking distributed across any surfaces, we would not see it as additional surface area. If much microcracking were present, it would make the energy deficit associated with the Griffith assumption much worse. But if Fig 4 is representative of the fragments, microcracking is not a major feature of Pyrex fracture.

It is significant that the data we use, to demonstrate that the Griffith assumption leads to an energy deficit

in Pyrex fragmentation, were derived independently by the writers and by Kolzenberg et al. (2013), although the latter did not note the significance of their data in the present context. The two sets of data yield the same result, which lends credibility to that result.

In the context of geological processes, our conclusion derived from Pyrex fragmentation applies also to intact rock (e.g. Handin et al., 1967; Kolzenberg et al., 2013), although the fracture surface energy varies with rock type and is affected by pre-existing structures (e.g. crystals and grains) that are absent in Pyrex. Davies et al. (*in press*) came to similar conclusions to the present work by constructing an energy budget

for a rock avalanche in sandstone material in New Zealand. They found that the maximum likely value for the potential energy released by the fall of the debris was very close to the minimum likely value for the energy lost to friction in the runout, so that a maximum of about 2% of potential energy was available to create new surface – far too little to explain the large area of new surface that was generated in the event. However, given that the SFE is a small proportion of FSE as Zgura et al. (2013) show for quartz,

the energy budget could be balanced satisfactorily. In this case consideration of the sub-micron fraction of the debris was critical, because it accommodated over 90% of the new surface area created in the rock avalanche, and its significance only became apparent when ultrasonic disaggregation of agglomerates was undertaken. We suspect that agglomerates form much more plentifully in the confined shearing of rock avalanche debris than in the unconfined compressive failure of Pyrex.

Our proposal that fracture surface energy transforms to elastic body-wave energy is well supported by the rock fracture literature. Since the 1970s researchers have recorded elastic body-waves in the form of acoustic emissions from crack propagation in brittle solids (e.g. Burnett, 2011), showing that extending cracks radiate energy. High-frequency (up to 0.6 MHz) acoustic emissions have been recorded prior to and during failure of rock samples (e.g. Wang et al., 2018; Schiavi et al., 2011; Carpinteri et al., 2012;

Michlmayr et al., 2012), confirming the existence of elastic body-waves associated with fracture. In fact

there is no literature claiming a lack of elastic strain release during brittle fracture. That crack-tip processes generate large quantities of very fine fragments at very high stresses has been confirmed by Reches and Dewers (2005). Although it is widely recognized that strain-energy is released by crack formation (Abraham, 2003), and that this energy is required to cause the breakage of further bonds thus

extending the crack with release of further bond energy, the fate of the released strain energy once a specific part of a crack extension is complete (and no further bonds are available to break) is not usually considered. In general, attention has hitherto focused on the micromechanics of energy transfer at the tip of a propagating crack, not on the later state and eventual fate of this energy. Our data, and our explanation of them, clarify this situation.

That such a fundamental assumption about the fate of fracture energy could remain unchallenged for so long (almost a century) appears peculiar at first sight. However, as mentioned earlier, the numerical difference to energy budgets resulting from the assumption only becomes significant when considering phenomena in which large numbers of sub-micron grains are formed, and when technologies are available to allow the finest fractions to be quantified. In Earth sciences the relevant phenomena are rock

avalanches, earthquakes, some explosive eruptions and bolide impacts, whose sedimentologies have only recently been investigated, while laser size analysis and electron microscope apparatus are also relatively recent arrivals on the particle-size distribution scene. This is demonstrated, as stated earlier, by the fact that in debris from a rock avalanche analysed by Davies et al (in press), over 90% of the surface area is associated with fragments less than 1 micron in diameter.

This result has far-reaching implications in a number of fields, particularly as technologies develop for identifying and measuring ever finer fragments. For example, estimates of input energy consumed as fracture surface energy in earthquakes (e.g. Cocco et al., 2006; Chester et al., 2005) and rock avalanches (e.g. Crosta et al., 2007) typically amount to a few percent, but with finer grains undoubtedly present in large quantities either as unrecognized agglomerates or below the available detection level these percentages could rise dramatically in future. In that case the energy remaining to radiate as seismic waves in earthquakes, or to explain the long runout of rock avalanches, would be severely underestimated if fracture surface energy were assumed to be a complete energy sink.

## 7. Conclusions

**7.1** Compressive failure of Pyrex cylinders consistently yields much more new surface area associated with fragments than is possible if fracture surface energy transforms to surface free energy as assumed by Griffith (1921).

**7.2** Empirical values for SFE from literature are in the order of 1-10% of FSE, again suggesting that the Griffith assumption is incorrect

**7.3** We suggest that most of the energy used to create new surface in brittle fracture transforms to elastic body-wave energy in the fragments, and is thus available to take part in further fracturing.

## 8. Author contribution

TRHD was responsible for experimental concept and design, data analyses and preparation of draft and final Mss.

MJM contributed the theoretical underpinning; carried out experiments and data acquisition; and contributed to draft and final Mss.

5 NVR contributed to experiments and data acquisition, especially particle-size distributions and SEM imaging.

**Acknowledgements**

The first author was partly supported by the Hazards Toolbox (T6) Of Resilience to Nature's Challenges, 10 funded by the Ministry of Business, Innovation and Employment, New Zealand, Grant Number RNC-011. Chris Grimshaw carried out some of the laser particle size analyses. Peter Jones designed and built the specimen container.

*Data availability.* Data are accessible by contacting Tim Davies by email (tim.davies@canterbury.ac.nz).

15 *Competing interests.* The authors declare that they have no conflict of interest.

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
