# Peer review of "What happens to Fracture Energy in Brittle Fracture? Revisiting the Griffith Assumption."

_Solid Earth, 2019_

## Referee Comment (RC1) · Anonymous Referee #1 · 12 Jun 2019

These experiments with Pyrex are interesting but there may be some weaknesses. 1) Only four tests are performed. The other data are taken from another source (Kolzenberg et al., 2013). 2) The kinetic energy of the fragments is not measured and thus energy considerations are not possible. The energy budget put into the system by compression should be the equal to the energy taken up by the fragments, by the acoustic, thermal energies, and by the energy used up to create new surface. Without estimating these forms of energy, it is difficult to conclude that fragments take up vibrational energy. In other words, one should demonstrate the presence of an energy deficit. 3) The figure in Fig. 8 shows the experimental data on the surface-energy plane. Three of the new data cluster at about 40 J. There is one isolated point at higher energy. This graph shows that more surface is created as consequence of energy loaded into the

system. If it were only for the data from this work, one would draw a fitting line between the four points. Adding the data from Kolzenberg et al. (2013) shows that data points are very much dispersed. It becomes difficult to find a fit. The general conclusion is that new surface approximately increases with energy, which is reasonable. 4) The energy per unit surface decreases as a function of the energy put into the rod (Table 3). It is possible that fragments acquire some internal energy as suggested here in addition to the kinetic, but this is still conjectural. 5) Application to geological phenomena is unclear, given the level of uncertainty. The paper would be strengthened with more tests. Moreover, other important forms of energy should be measured and/or better constrained.

---

## Referee Comment (RC2) · Timothy Davis (Referee) · 24 Jun 2019

General Comments

This article focuses on fracture surfaces created in Pyrex glass cylinders subject to confined (and unconfined) compression. Both new results and previous results from Kolzenberg (2013) are used in the analysis. Assuming these Pyrex cylinders have material properties as described in the literature (surface potential energy) then the authors conclude the amount of surface area created in the experiments exceeds the amount that would predicted from early material failure theory. The article sets up the problem, describes the experiments and interprets the results. Some broad statements are made suggesting Griffith theory (1921) may be invalid, inferred from these experimental results. To aid such statements I suggest the authors outline the theory with equations clearly in the introduction chapter to show it is understood and provide an explanation later in the manuscript some reasons as to why this long-standing assumption has previously prevailed for so long.

Major revision is suggested, the results are interesting but clarification of some assumptions and additional details on the setup must be provided in certain parts of the manuscript to validate the interpretation.

Specific Comments

- The abstract attempts to describe the paper but is more leading than clearly describing the setup and results that test the quoted long-standing assumptions concisely. It also conclusively suggests that most of the energy is radiated from the tip as elastic wave energy but no data in the article is provided to validate this statement, this is only postulated later in the discussion section. I suggest the abstract is rewritten such that both the experiments/results are summarised, not just the interpretation of the results. The experimental procedure employed in this work is suitable, well documented (both setup and results) and clearly thought out. A discussion on the limitations and potential error introduced while calculating the fracture surface calculations is well documented and appropriate. Some assumptions/questions:

- In chapter Machine strain energy and specimen strain energy I am unsure to whether there is enough detail on the mechanical system. In chapter 9.2.1 Pollard and Fletcher, Fundamentals of structural geology (2005) machines can be categorised as a 'stiff or 'soft' testing machines, relative to the sample. This definition includes the energy stored in machines frame and does not only concentrate on the material properties of parts of the machine in contact with the sample and as such, is an important detail. Some evidence that the post failure behaviour of this setup is stable is desired. Without such discussion how can readers be certain about the energies in the system that are discussed?

- How confident are the authors on stating a single value for the empirically derived free surface energy criteria of Pyrex described in Wiederhorn (1969)/Lange (1971). For example, Table 2 of Wiederhorn gives a standard deviation of $\sim \pm$ 5% for borosilicate glass fracture surface energy and Lange also gives a $\pm$ 5% scatter. Some remarks on potential variations in the literatures empirical value of this properly with some estimated standard deviation or maximal/minimal values would be more suitable and figure 8 should be updated to show this. Following on from this, is there evidence the glasses used in the experiment have the same values as those in the literature? Note the detailed annealing setup to create identical thermal history of the glass slides used in Wiederhorn (1969) is different. It would help to add some discussion of how well the material properties of the glasses used match/deviate from those detailed the literature.

- It is of interest to discuss if the findings of this study only work for solids in compression. If not, discuss why previously estimated surface energies of previous studies such as Wiederhorn (1969) and Lange (1971) match so well?

- Chapter 10.7 & 10.8: Jaeger, J.C., Cook, N.G. and Zimmerman, R., 2009. Fundamentals of rock mechanics. John Wiley & Sons. Here it is stated:

"Irwin (1958) extended Griffith's concept by pointing out that in many materials, as a crack grows, energy must also be expended to create a damaged zone of irreversible, plastic deformation ahead of the crack tip. In rock, this zone may consist of crushed grains, microcracking, etc."

Do the authors believe there is a way to differentiate such micro cracking in their particle diameter distributions (Fig 3) and would this account for the additional fracture surface they see? It must also be noted that the experiments of Wiederhorn and Lange calculate the fracture surface energy based on the propagation length of a pre-existing crack at the scale of the sample macroscale sample (75 by 25mm/ 300 by 150 mm). No focus in these studies is given to estimating micro cracking (if there was any) close to the crack plane/tip, as such are these empirical values valid for the scales used in

this study?

- Two of the references (Hungr/Livne) in chapter 2 page 3 line 17/18 seem irrelevant to the statement in hand (free surface energy is treated as an energy sink):

Reference 1 (Miller): This is fine, the article details that free surface energy is an energy sink, tensional numerical experiments.

Reference 2 (Chester): This is fine, estimation of free surface energy in fault gouge.

Reference 3 (Hungr): No reference to free surface energy or the energy sink in this publication, it deals with. Frictional experiments.

Reference 4 (Grady): This is fine, the article states "Fracture in brittle solids is weakly dissipative in the sense of converting available energy into surface energy or plastic work. Thus, failure through one or several through-going cracks is far from adequate for absorbing the initial stored elastic strain energy. Consequently, during failure, fracture on successively finer length scales proceeds through a cascade of crack branching until a length scale adequate to the dissipation of the initial elastic strain energy is achieved."

Reference 5 (Livne): I am not sure this this is appropriate for the point being made, firstly they are not using free surface energy in the publication, instead this focusses on testing how well linear elasticity applies and how plasticity (or at least non-linear elastic deformation) at the tip is an energy sink. Correct me if I am wrong but from what I understand basic Griffith failure theory is not directly being used here.

Technical corrections

In introduction page 2 line 9: change 'gravity' to 'gravitational'.

In introduction page 3 line 4 change 'to understanding' to 'to the understanding'.

In Machine strain energy and specimen strain energy line 14 change 'failure of cylinder' to 'failure of the cylinder'.

---

## Editor Comment (EC1) · David Healy (Editor) · 19 Jul 2019

Dear Tim

thanks for responding to the reviewer comments. I have read through your replies.

Please now proceed to prepare and submit a revised manuscript. I would ask you to specifically address the detailed comments by reviewer Tim Davis, and to clarify the main thrust of your paper (which has caused confusion for both reviewers).

When we receive the new ms, we will decide on the next steps, which may involve further review(s).

All the best, Dave

---

## Author Response (AR1)

**Response**

**Introduction to Response:**

*We believe the Referee may have misconstrued the objective of our manuscript.*

*Our purpose is to demonstrate that the area of new surface resulting from a brittle fragmentation event exceeds that possible if the Griffith assumption, that fracture energy always instantly transforms to unavailable surface energy, is valid. This demonstration ideally requires only a single data point, if the accuracy of that point is solidly established; we have chosen to report four of our data points that demonstrate the invalidity of the assumption, and further corroborate these with **independent** data from another source (Kolzenberg et al., 2013). These ten data points make an extremely strong case for the lack of universal validity of the Griffith assumption. In fact we also conducted experiments that resulted in the creation of fewer (in one case only two) fragments, and so did not invalidate the Griffith assumption; since these data, however, do not affect the reality of those that do invalidate the assumption, we did not report them.*

*Had our objective been to report on the pattern of surface area generation by failures at a range of stresses, then certainly more data would have been needed; but they were not.*

*Changes in manuscript: We have added the following sentence to the Discussion to emphasise the above response (p. 27 lines 3-5):*
"While the data presented are widely scattered (Fig. 8), this is irrelevant to the specific objective of our work, which is to demonstrate that the new surface area generated in brittle fracture is not limited to the zone below the dashed lines in Fig. 8, as it would be if the Griffith assumption were valid."

*Introductory comment: "*These experiments with Pyrex are interesting but there may be some weaknesses."

*Response: We assume that the referee's numbered comments refer to the weaknesses. We reproduce the comments and respond to them individually*

*Changes in manuscript:* None

**Comment:** "1) Only four tests are performed. The other data are taken from another source (Kolzenberg et al., 2013)."

**Response**: *As noted in our introduction above, if the objective of the research is understood this is not a weakness. Certainly, independent corroborative data constitute a strength.*

**Changes in manuscript:** None

**Comment**: "2) The kinetic energy of the fragments is not measured and thus energy considerations are not possible. The energy budget put into the system by compression should be the equal to the energy taken up by the fragments, by the acoustic, thermal energies, and by the energy used up to create new surface. Without estimating these forms of energy, it is difficult to conclude that fragments take up vibrational energy. In other words, one should demonstrate the presence of an energy deficit."

**Response**: *Fragment kinetic energy had reduced to zero by the time we removed the shattered sample from the container. Our energy budget compares the elastic strain energy immediately prior to fragmentation with the surface energy when the fragmentation episode had finished and all the transitional energies had dissipated. The fragment kinetic energy present during the fragmentation event would have transformed to surface energy as moving fragments further fragmented, and to elastic, thermal and acoustic energies upon impact with other fragments and with the container walls; these energies may or may not have contributed to further fragmentation. We have not measured the elastic, thermal and acoustic energies emitted during the test that did not cause fragmentation; if we had done so, these would have had positive values, so the energy available to create new surface would have been less than we have assumed. The energy deficit we have demonstrated would thus have been larger, further supporting our conclusion.*
*Our data throughout the analysis have been chosen to be consistently conservative, that is, to be as detrimental to our outcome as possible. Hence the large uncertainty of the data do not detract from our definite conclusion*

**Changes in manuscript:** None

**Comment:** "3) The figure in Fig. 8 shows the experimental data on the surface-energy plane. Three of the new data cluster at about 40 J. There is one isolated point at higher energy. This graph shows that more surface is created as consequence of energy loaded into the system. If it were only for the data from this work, one would draw a fitting line between the four points. Adding the data from Kolzenberg et al. (2013) shows that data points are very much dispersed. It becomes difficult to find a fit. The general conclusion is that new surface approximately increases with energy, which is reasonable."

**Response**: *While it is not our objective to establish a relationship between energy available and new surface created, the Griffith theory requires that there is a positive relationship*

*which, as the reviewer points out, appears to be the case. However this is not directly relevant to the objective of our research as pointed out above - the invalidity of the Griffith assumption about surface energy, which is unconnected with the relationship between energy and surface area.*

***Changes in manuscript:*** None

**Comment:** "4) The energy per unit surface decreases as a function of the energy put into the rod (Table 3). It is possible that fragments acquire some internal energy as suggested here in addition to the kinetic, but this is still conjectural."

**Response**: *energy per unit surface is the gradient of the data in Fig. 8. The trend suggested by the reviewer from Table 3 thus implies that the local gradient reduces as energy input increases – this is not apparent in Fig. 8. Again, identifying such trends is not the objective of the work.*

***Changes in manuscript:*** None

**Comment:** "5) Application to geological phenomena is unclear, given the level of uncertainty. The paper would be strengthened with more tests. Moreover, other important forms of energy should be measured and/or better constrained."

**Response**: *As set out in our Introduction to Responses, there is NO uncertainty in our interpretation of the data. It is unequivocal that all the data contradict the Griffith assumption. Consideration of other forms of energy would, unless they are negative, strengthen our conclusion.*

***Changes in manuscript:*** None
We are grateful to Davis for his detailed comments, to which we respond as follows:

**General comments**

**Comment** "Some broad statements are made suggesting Griffith theory (1921) may be invalid".

**Response**: *We wish to emphasise strongly that this is not a correct summary of our manuscript. In section 2 we have written "Whether or not this assumption is true has no effect on the validity of the Griffith fracture theory, which only addresses the mathematical requirements for a crack to enlarge – i.e. the value of FSE. The theory is not about SFE and says nothing about the complex chemical processes associated with exposure of fresh material by enlargement of cracks." This makes it clear that we are questioning* **only** *the Griffith (1921)* **assumption** *about the fate of elastic strain energy once the crack has been extended,* **not** *the theory itself.*

**Changes in manuscript:** We have amended the Introduction to the following (new text in yellow highlight; p2 line 5 to p3 line 11):

**"1 Introduction**

[revised manuscript text omitted]

**Comment:** "the authors outline the theory with equations clearly in the introduction chapter to show it is understood";

**Response**: *because we are not questioning anything within the theory itself such a detailed outline would not be relevant to our arguments, and might mislead readers. We believe our rewriting of the Introduction, together with the description in section 2, address this point adequately.*

**Changes in manuscript:** None

**Comment:** "provide an explanation later in the manuscript some reasons as to why this long-standing assumption has previously prevailed for so long."

 **Response**: *We have done this.*

**Changes in manuscript:**
Penultimate paragraph of section 6 (p. 29 line 10- 19):
That such a fundamental assumption about the fate of fracture energy could remain unchallenged for so long (almost a century) appears peculiar at first sight. However, as mentioned earlier, the numerical difference to energy budgets resulting from the assumption only becomes significant when considering phenomena in which large numbers of sub-micron grains are formed, and when technologies are available to allow the finest fractions to be quantified. In Earth sciences the relevant phenomena are rock avalanches, earthquakes, some explosive eruptions and bolide impacts, whose sedimentologies have only recently been investigated, while laser size analysis and electron microscope apparatus are also relatively recent arrivals on the particle-size distribution scene. This is demonstrated, as stated earlier, by the fact that in debris from a rock avalanche analysed by Davies et al (in press), over 90% of the surface area is associated with fragments less than 1 micron in diameter.

**Specific Comments**

**Comment:** "The abstract attempts to describe the paper but is more leading than clearly describing the setup and results that test the quoted long-standing assumptions concisely. It also conclusively suggests that most of the energy is radiated from the tip as elastic wave energy but no data in the article is provided to validate this statement, this is only

postulated later in the discussion section. I suggest the abstract is rewritten such that both the experiments/results are summarised, not just the interpretation of the results."

**Response**: *We are very happy to follow this useful and constructive suggestion.*

***Changes in manuscript:*** **(**P.1 lines 7-19)

**Abstract.** Laboratory experiments involving unconfined compressive failure of borosilicate glass cylinders quantified the elastic strain energy released at failure and the size distribution of the resulting fragments. The data were carefully assessed for potential inaccuracies in surface-area calculation, of the contribution of energy from the compression machine relaxation during specimen failure, and of possible variations in the specific fracture energy of the specimens. The data showed that more new surface area was created during the failures than would be possible if the long-standing assumption, that all the energy involved in creating new rock surface area in brittle material is taken up by the newly-created surfaces as surface potential energy and is unavailable to do further work, were valid. We therefore conclude that the assumption is false. This conclusion is supported by independent data from a previous investigation whose authors did not pursue this particular application. Our result does not affect the validity of Griffith fracture mechanics, and is significant only when large numbers of very fine fragments are created by brittle fracture, as in rock-avalanche motion and earthquake rupture, and are identified in particle-size distributions. In such situations our result is very significant to understanding of fracture energetics.

**Comment**: "In chapter Machine strain energy and specimen strain energy I am unsure to whether there is enough detail on the mechanical system. In chapter 9.2.1 Pollard and Fletcher, Fundamentals of structural geology (2005) machines can be categorised as a 'stiff or 'soft' testing machines, relative to the sample. This definition includes the energy stored in machines frame and does not only concentrate on the material properties of parts of the machine in contact with the sample and as such, is an important detail. Some evidence that the post failure behaviour of this setup is stable is desired. Without such discussion how can readers be certain about the energies in the system that are discussed?"

**Response**: *The testing machine is commercially marketed as having a stiff frame. The only relevance of the post-failure behaviour of the setup is the energy that can enter the fracturing body during its failure due to rebound of the machine frame during this failure. This very topic is discussed in sufficient detail in section 3.3 of the manuscript to demonstrate that the contribution of machine energy to the failure process of the specimen is small – and certainly not large enough to affect the energy budget to such an extent that our questioning of the Griffith assumption is called into question.*

***Changes in manuscript:*** We have emphasised the time limitation of the machine energy contribution by

(i) adding to p. 8 lines 10-11: "This machine is marketed as having a stiff frame."

(ii) adding to the sentence ending on line 3 of p. 9: "during the failure of the specimen".

(iii) adding to p. 10 lines 10-13: "While the whole of the machine frame stores elastic strain energy during specimen compression, only that released during specimen failure can contribute to the energy available to create new fragment surface." and omitted "Further" from the start of the following sentence.

**Comment**: "How confident are the authors on stating a single value for the empirically derived free surface energy criteria of Pyrex described in Wiederhorn (1969)/Lange (1971). For example, Table 2 of Wiederhorn gives a standard deviation of ±5% for borosilicate glass fracture surface energy and Lange also gives a ±5% scatter. Some remarks on potential variations in the literatures empirical value of this properly with some estimated standard deviation or maximal/minimal values would be more suitable and figure 8 should be updated to show this. Following on from this, is there evidence the glasses used in the experiment have the same values as those in the literature? Note the detailed annealing setup to create identical thermal history of the glass slides used in Wiederhorn (1969) is different. It would help to add some discussion of how well the material properties of the glasses used match/deviate from those detailed the literature."

**Response**: *We note firstly that altering the value of specific fracture energy of borosilicate glass has negligible effect on our data and conclusions. We have amended Fig. 8 to show a ± 5% variation in fracture energy, and it makes no difference at all to the significance of the plot and added a paragraph to section 4 Data explaining this.*

*Commercial and scientific data are unanimous in stating that the fracture energy of borosilicate glass is close to 4.5 Jm$^{-2}$, and Kolzenberg et al (2013) also adopted this value; we emphasise again the fact that our data are similar to those of Kolzenberg et al (2013), so it is most unlikely that our specimens were in any way unrepresentative of borosilicate glasses.*

**Changes in manuscript:**
We have amended the dashed line in Fig. 8 to represent a possible ±5% variation in fracture energy of Pyrex, from 4.28 Jm$^{-2}$ to 4.72 Jm$^{-2}$, and amended the figure caption correspondingly.

We have also altered the final column of Table 3 to show the maximum possible surface area under the Griffith assumption with the lowest possible value of U = 4.28 Jm$^{-3}$.

We have added the following to p.20 lines 7-10:
"Similarly the effect of possible variations in the specific fracture energy of borosilicate glass (4.5 ± 0.22 Jm$^{-2}$; Wiederhorn, 1969) are included in the location of the dashed line in Fig. 8 and have negligible effect. The theoretical maximum surface area corresponding to the energy available are also estimated (Table 1 final column) assuming that the specific fracture energy is 4.28 Jm$^{-2}$."

**Comment**: "It is of interest to discuss if the findings of this study only work for solids in compression. If not, discuss why previously estimated surface energies of previous studies such as Wiederhorn (1969) and Lange (1971) match so well?"

**Response**: *While the stresses applied to the specimens were compressive, the resultant stress fields within the specimen leading to and at failure would not have been purely compressive. It is well-known, for example, that diagonal shear stress concentrations occur in cylinders under axial compression. What is important to our experiments is the quantity of elastic strain energy stored in the specimen at failure; because different applied stress geometries generate different internal stress fields they will also result in different stored energies and thus different fragment size distributions.*

*Nevertheless, even if our chosen setup is particularly suited to creating large numbers of small fragments, this does not alter the conclusion that the Griffith assumption was violated in our case. This is sufficient evidence that the assumption – which was inferred by Griffith (1921) to apply universally, and has since been held to apply universally – is false.*

**Changes in manuscript:** We have added the following to the Discussion (p. 26 lines 17-19, p. 27 lines 1-2):
"While our data were acquired under specific conditions of system geometry, material and applied stress configuration, which may perhaps have been particularly favourable for reaching our outcome, our demonstration that the Griffith assumption is not valid in these circumstances is a very strong indicator that it is invalid in all circumstances. Griffith's (1921) original assumption was associated with a very general derivation of his theory, and therefore implies general applicability of both theory and assumption."

**Comment:** "Chapter 10.7 & 10.8: Jaeger, J.C., Cook, N.G. and Zimmerman, R., 2009. Fundamentals of rock mechanics. John Wiley & Sons. Here it is stated: "Irwin (1958) extended Griffith's concept by pointing out that in many materials, as a crack grows, energy must also be expended to create a damaged zone of irreversible plastic deformation ahead of the crack tip. In rock, this zone may consist of crushed grains, micro-cracking, etc."
Do the authors believe there is a way to differentiate such micro cracking in their particle diameter distributions (Fig 3) and would this account for the additional fracture surface they see? It must also be noted that the experiments of Wiederhorn and Lange calculate the fracture surface energy based on the propagation length of a pre-existing crack at the scale of the sample macroscale sample (75 by 25mm/ 300 by 150 mm). No focus in these studies is given to estimating micro cracking (if there was any) close to the crack plane/tip, as such are these empirical values valid for the scales used in this study?"

**Response**: *The Irwin (1958) extension assumes irreversible plastic deformation. In cracking of Pyrex glass, if there were crushed-but-not-disaggregated grains, or micro-fractures in the surviving fragments, they represent additional fracture surface area created that would not be accounted for in our calculation of apparent surface area from the particle-size distributions. That is, if there is micro-cracking distributed across any surfaces, we would not*

*see it as additional surface area. Hence, if much micro-cracking were present, it would only make the problem much worse – that is, the true energy budget would be even more in deficit. But if our Fig 4 is representative of the fragments, micro-cracking is not a major feature (although it is present).*

***Changes in manuscript:*** We added the following to the Discussion (p. 27 lines 6-13): "We note that we have ignored the possibility that plastic deformation occurred in the glass failure. Irwin (1958) assumed that irreversible plastic deformation can occur. If in cracking of Pyrex glass if there were crushed-but-not-disaggregated grains, or microfractures in the surviving fragments, they would represent additional fracture surface area created that would not be accounted for in our calculation of surface area from the particle-size distributions. That is, if there is microcracking distributed across any surfaces, we would not see it as additional surface area. If much microcracking were present, it would make the energy deficit associated with the Griffith assumption much worse. But if Fig 4 is representative of the fragments, microcracking is not a major feature."

***Comment:*** It must also be noted that the experiments of Wiederhorn and Lange calculate the fracture surface energy based on the propagation length of a pre-existing crack at the scale of the sample macroscale sample (75 by 25mm/ 300 by 150 mm). No focus in these studies is given to estimating micro cracking (if there was any) close to the crack plane/tip, as such are these empirical values valid for the scales used in this study?"

***Response:*** *Davis raises the question of scale dependence of borosilicate properties, but fracture surface energy is not supposed to be a scale-dependent property for any material.*

***Changes in manuscript:*** None

***Comment***: "Reference 3 (Hungr): No reference to free surface energy or the energy sink in this publication, it deals with. Frictional experiments.

***Response***: *The reference Hungr and Morgenstern, 1984 is not cited in the text and will be deleted. However the reference Hungr, 2006 does refer to rock fragmentation as an energy sink and is retained.*

***Changes in manuscript:*** Delete reference Hungr and Morgenstern (1984)

***Comment***: "Reference 5 (Livne): I am not sure this this is appropriate for the point being made, firstly they are not using free surface energy in the publication, instead this focusses on testing how well linear elasticity applies and how plasticity (or at least non-linear elastic deformation) at the tip is an energy sink. Correct me if I am wrong but from what I understand basic Griffith failure theory is not directly being used here."

***Response***: *Davis is correct. However, as stated above, it is not basic Griffith theory that is the topic of our manuscript, it is the fate of energy involved in extending a crack in brittle material once the new crack has been created. The relevance of Livne et al. (2010) in the*

*present context is simply that it adopts the Griffith assumption that the energy is dissipated at the crack tip as the crack extends.*

***Changes in manuscript:*** None

***Comment***: "Technical corrections
In introduction page 4 line 2: change 'gravity' to 'gravitational'.
In introduction page 5 line 4change 'to understanding' to 'to the understanding'.
In Machine strain energy and specimen strain energy (p. 10 line 14) change 'failure of cylinder'
to 'failure of the cylinder'."

***Response***: *We are happy to adopt all these recommendations*

***Changes in manuscript:***
- In introduction page 4 line 2: we have changed 'gravity' to 'gravitational'.
- In introduction page 5 line 4 we have changed 'to understanding' to 'to the understanding'.
- In Machine strain energy and specimen strain energy P 10 line 14 we have changed 'failure of cylinder' to 'failure of the cylinder'."

***Other changes in manuscript:***

Davies et al. (in review) in now "(in press)" p. 4 line 3, p. 28 line 4 and "In press, Landslides.*"* p. 33 line 14